# Spatiotemporal Gene Expression Regions along the Anterior–Posterior Axis in Mouse Embryos before and after Palatal Elevation

**DOI:** 10.3390/ijms23095160

**Published:** 2022-05-05

**Authors:** Arata Nagasaka, Koji Sakiyama, Yasuhiko Bando, Masahito Yamamoto, Shinichi Abe, Osamu Amano

**Affiliations:** 1Division of Histology/Anatomy, Meikai University School of Dentistry, 1-1 Keyakidai, Sakado 350-0283, Japan; sakiyama@dent.meikai.ac.jp (K.S.); y-bando@dent.meikai.ac.jp (Y.B.); oamano@dent.meikai.ac.jp (O.A.); 2Department of Anatomy, Tokyo Dental College, 2-9-18, Kandamisaki-cho, Chiyoda-ku, Tokyo 101-0061, Japan; yamamotomasahito@tdc.ac.jp (M.Y.); abesh@tdc.ac.jp (S.A.)

**Keywords:** mouse, palate development, palatal shelf elevation, Pax9, Osr2

## Abstract

The mammalian secondary palate is formed through complex developmental processes: growth, elevation, and fusion. Although it is known that the palatal elevation pattern changes along the anterior–posterior axis, it is unclear what molecules are expressed and whether their locations change before and after elevation. We examined the expression regions of molecules associated with palatal shelf elevation (Pax9, Osr2, and Tgfβ3) and tissue deformation (F-actin, E-cadherin, and Ki67) using immunohistochemistry and RT–PCR in mouse embryos at E13.5 (before elevation) and E14.5 (after elevation). Pax9 was expressed at significantly higher levels in the lingual/nasal region in the anterior and middle parts, as well as in the buccal/oral region in the posterior part at E13.5. At E14.5, Pax9 was expressed at significantly higher levels in both the lingual/nasal and buccal/oral regions in the anterior and middle parts and the buccal/oral regions in the posterior part. Osr2 was expressed at significantly higher levels in the buccal/oral region in all parts at E13.5 and was more strongly expressed at E13.5 than at E14.5 in all regions. No spatiotemporal changes were found in the other molecules. These results suggested that Pax9 and Osr2 are critical molecules leading to differences in the elevation pattern in palatogenesis.

## 1. Introduction

Craniofacial development is a complex, multistep morphological process. This process is spatiotemporally regulated by numerous molecular mechanisms [1]. Cranial neural crest cells are regulated by N-cadherin signaling for directional cell migration during embryonic development, thus contributing to the formation of each structure, such as cartilage, bone, smooth muscle, and many others, in the particular location at which they arrive [2,3]. Similarly, the organizers that supply morphogens for craniofacial morphogenesis are located in certain areas of the facial ectoderm [4]. Disruption of these molecular mechanisms results in craniofacial malformation [5]. In the oral region, it has been shown that paired-box gene 9 (Pax9) is important for maintaining the expression of genes such as msh homeobox 1 (MSX1) and lymphoid enhancer-biding factor 1 (LEF1), and these defects have been associated with a lack of tooth buds and hypodontia [6,7]. Moreover, one of the most common congenital anomalies of the craniofacial region is cleft palate [8].

The secondary palate is formed by an ordered sequence of events: growth, elevation, and fusion. Palate development starts from the appearance of palatal primordia at the lateral edges of the maxillary process. The bilateral palatal shelves subsequently grow down vertically along the sides of the tongue. Then, the palatal shelves undergo a process called palatal shelf elevation, in which the vertical palatal shelves orient horizontally above the tongue from embryonic day 13.5 (E13.5) to E14.5 in mice. After palatal elevation, the palatal shelves grow toward the midline and fuse with each other to form a continuous palate [9,10]. Previous studies have shown that palate development is mediated by multiple molecules [11,12,13]. Studies of mutant mice revealed that paired-box-gene-9 (Pax9)-deficient, odd-skipped-related-transcription-factor-2 (Osr2)-null, and transforming growth factor beta3 (Tgfβ3) mice exhibit complete cleft palate [6,14,15]. In addition, these molecules have been reported to cooperate with each other and contribute to normal palatal development [10,12].

We focused on palatal shelf elevation. Previous studies suggested that the elevation pattern differs along the anterior–posterior axis of the palatal shelf. One is the “flip-up” model in the anterior (coronal plane anterior to the molar tooth bud) and middle (coronal plane identical to the molar tooth bud) parts, in which the palate shelves move from lateral to above the tongue and change their orientation in the oral cavity from vertical to horizontal; the other is the “flow” model in the posterior (coronal plane posterior to the molar tooth bud) part, in which the horizontal palate shelves are formed by outgrowth from the side of the vertical palate shelves and flow over the tongue [16,17]. These different elevation patterns suggest the involvement of region-specific molecules. The expression patterns of Pax9, Osr2, and Tgfβ have been analyzed through in situ hybridization [15,18,19,20]; however, those data cannot be simply compared due to differences in the genetic backgrounds and the lack of analysis regions. Therefore, we examined the expression region of molecules associated with palatal shelf elevation and tissue deformation in mouse embryos with the same genetic background. To perform a more detailed spatiotemporal analysis, we examined the lingual/nasal and buccal/oral regions of the palatal shelf in E13.5 and E14.5 mouse embryos before and after elevation separately for the three parts along the anterior–posterior axis of the palate: anterior, middle, and posterior. In each of these regions, we examined the expression of Pax9, Osr2, and Tgfβ3 in relation to palatal shelf elevation and that of F-actin, E-cadherin, and Ki67 in relation to tissue deformation.

## 2. Materials and Methods

### 2.1. Animals

Pregnant ICR mice were obtained from Sankyo Labo Service (Tokyo, Japan). Embryonic day zero (E0.5) was defined as the day of vaginal plug identification. We chose E13.5 at 12:00 a.m. in mouse embryos in which the palatal shelves grew down vertically along the sides of the tongue and palatal elevation had not yet occurred, and we chose E14.5 at 12:00 a.m. in mouse embryos in which palatal elevation had been finished.

This study was approved by the animal ethics committee, as described in the Institutional Review Board Statement section.

### 2.2. Immunofluorescence

E13.5 and E14.5 embryo heads were fixed in 4% paraformaldehyde overnight, immersed for 24 h in 30% sucrose, embedded in OCT compound (Sakura Finetek Japan, Tokyo, Japan), frozen, and sectioned coronally (16 μm thick). Frozen sections were treated with the following primary antibodies: anti-Pax9 (rat, sc-56823, Santa Cruz Biotechnology, 1/300, Dallas, TX, USA), anti-Osr2 (mouse, sc-393516, Santa Cruz Biotechnology, 1/100), anti-TGF beta 3 (rabbit, ab15537, Abcam, 1/1000, Cambridge, UK), anti-E-cadherin (rabbit, 20874-1-AP, Proteintech, 1/400, Wuhan, Hubei), and anti-Ki67 (rabbit, NB500-170, Novus Biologicals, 1/400, Littleton, CO, USA). After washing, the sections were treated with secondary antibodies conjugated with Alexa Fluor 488 (A11034, A11006, Invitrogen, 1/200, Waltham, MA, USA) or Phalloidin-iFluor 488 conjugate (20549, Cayman Chemicals, 1/500, Ann Arbor, MI, USA). Confocal images were obtained with an LSM800 confocal laser microscope (Carl Zeiss, Obercohen, Germany) equipped with Zen 2.1 software (Carl Zeiss). The objective lens used was a Plan-Apochromat 20×/0.8 M27.

### 2.3. Preparation of Palatal Shelf Samples for Real-Time PCR

To measure region-specific differences, the palatal shelf was freshly isolated from embryos at E13.5 and E14.5 and processed microsurgically in DMEM/F12 (05177-15, Nacalai Tesque, Kyoto, Japan). The palatal shelf was sectioned into anterior, middle, and posterior parts and then separately sectioned into lingual/nasal and buccal/oral regions (*n* = 5 mouse embryos for each region).

### 2.4. Quantitative Real-Time PCR

Total RNA was extracted by using an RNeasy Micro Kit (Qiagen, Hilden, Germany) according to the manufacturer’s instructions. RNA was reverse-transcribed with a Transcriptor First Strand cDNA Synthesis Kit (Roche, Basel, Switzerland). Quantitative PCR amplifications were performed in a LightCycler 480 (Roche) using the LightCycler 480 SYBR GreenIMaster Mix (Roche). Primer pairs for Pax9-F (5′-CAGCAGCTAAGGTGCCTACA-3′) and Pax9-R (5′-CTGTCGCTCACTCCTTGGTC-3′), Osr2-F (5′-ACCAATTACCGCTGTCGCTT-3′) and Osr2-R (5′-ACAACAGCACGCAGAGGAAT-3′), Tgfβ3-F (5′-ACTGGCGGAGCACAATGAA-3′) and Tgfβ3-R (5′-GTGCTCATCCGGTCGAAGTA-3′), Cadherin-F (5′-GGCTGGACCGAGAGAGTTAC-3′) and Cadherin-R (5′-TGTGCTCAAGCCTTCACCTT-3′), Actin-F (5′-ACAGAGAGAAGATGACGCAGATAA-3′) and Actin-R (5′-CATGACAATGCCAGTGGTGC-3′), Ki67-F (5′-AAGACAATCATCAAGGAACGCC-3′) and Ki67-R (5′-ATGGATGCTCTCTTCGCAGG-3′), and glyceraldehydo-3-phosphate dehydrogenase (GAPDH)-F (5′-GGTTGTCTCCTGCGACTTCA-3′) and GAPDH-R (5′-GCCGTATTCATTGTCATACCAGG-3′) were obtained from Eurofins Genomics (Tokyo, Japan). The expression levels of the target genes in each sample were normalized to the GAPDH levels. The results are expressed as the fold change in gene expression compared to the lingual/nasal regions and E13.5 as a control, and they are presented as the mean ± SEM. Student’s *t*-test was used to analyze the difference, and *p* < 0.05 and *p* < 0.01 were considered statistically significant.

## 3. Results

### 3.1. Spatiotemporal Expression Location of Molecules Related to Palatal Shelf Elevation

To investigate the localization of molecules during palatal shelf elevation, we observed both the lingual/nasal and buccal/oral regions of the palatal shelf in E13.5 and E14.5 mouse embryos separately for the three parts along the anterior–posterior axis: anterior (anterior to the molar tooth bud), middle (same plane as the molar tooth bud), and posterior (posterior to the molar tooth bud) (Figure 1). We first examined the expression regions of molecules related to palatal shelf elevation and tissue deformation through immunohistochemistry.

Pax9 is a member of the transcription factor family, which is characterized by the paired-class DNA-binding domain [21]. Pax9 was more strongly expressed in the lingual/nasal region than in the buccal/oral region in the anterior and middle parts of the palatal mesenchyme at E13.5. In contrast, in the posterior part of the palatal mesenchyme at E13.5, Pax9 was more strongly expressed in the buccal/oral region than in the lingual/nasal region (Figure 2A–C). At E14.5, Pax9 was expressed throughout the palatal mesenchyme in the anterior parts. In the middle and posterior parts, Pax9 was expressed in the palatal mesenchyme near the fusion site. In addition, as in E13.5, the buccal/oral region expression was stronger than the lingual/nasal region expression in the posterior part (Figure 2D–F).

Osr2 is a mammalian homolog of the *Drosophila* odd-skipped family developmental regulators [22,23]. Osr2 was expressed throughout the palatal mesenchyme from anterior to posterior at E13.5, with the lingual/nasal regions expressing more strongly than the buccal/oral regions (Figure 2G–I). In E14.5, Osr2 expression was reduced throughout the palatal shelf in the anterior to posterior direction compared to E13.5 (Figure 2J–L).

Tgfβ3 is a member of a large family of cytokines called the transforming growth factor beta superfamily [24]. Tgfβ3 was expressed at both E13.5 and E14.5 in the anterior-to-posterior epithelium, and it was equally expressed in the lingual/nasal and buccal/oral regions (Figure 2M–R).

### 3.2. Spatiotemporal Expression Location of Molecules Related to Tissue Deformation

We next examined the expression regions of molecules related to tissue deformation. Fluorescently labeled phalloidin is a marker for the cytoskeletal F-actin network. F-actin was expressed throughout the epithelium and mesenchyme from the anterior to posterior palatal shelf at both E13.5 and E14.5. It was also expressed to the same extent in the lingual/nasal and buccal/oral regions (Figure 3A–F). Cadherin is a family of glycoproteins involved in the Ca^2+^-dependent cell–cell adhesion mechanism [25]. E-cadherin was expressed at a similar location to that of Tgfβ3 in the anterior-to-posterior epithelium and was equally expressed in the lingual/nasal and buccal/oral regions at both E13.5 and E14.5 (Figure 3G–L). Ki67 is a marker of cell proliferation and the cell cycle. Ki67 was widely expressed in the epithelium and mesenchyme from the anterior to posterior palatal shelf and equally expressed in the lingual/nasal and buccal/oral regions (Figure 3M–R).

### 3.3. Spatial Differences in the Gene Expression Locations of Molecules

We observed locational molecular changes through immunohistochemistry before and after palatal shelf elevation. To quantify the extent to which there were differences in expression, we performed real-time PCR. First, to clarify spatial differences, we compared gene expression levels between the lingual/nasal and buccal/oral regions in the palatal shelf at E13.5 (Figure 4A–F). Pax9 gene expression was significantly higher in the lingual/nasal region than in the buccal/oral region, both in the anterior and middle parts. In contrast, the buccal/oral region expression in the posterior part was significantly higher than that in the lingual/nasal region (Figure 4A). Osr2 gene expression was significantly higher in the buccal/oral region than in the lingual/nasal region in the anterior, middle, and posterior parts (Figure 4B). There was no significant difference in the gene expression of Tgfβ3, F-actin, E-cadherin, or Ki67 in the lingual/nasal and buccal/oral regions (Figure 4C–F).

On the palatal shelf at E14.5, Pax9 gene expression was not significantly different in the anterior and middle parts, but was significantly higher in the buccal/oral region than in the lingual/nasal region in the posterior part, as in E13.5 (Figure 4G). In contrast to E13.5, Osr2 gene expression was not significantly different between the lingual/nasal and buccal/oral regions (Figure 4H). Tgfβ3, F-actin, E-cadherin, and Ki67 gene expression was not significantly different in the lingual/nasal and buccal/oral regions, as in E13.5 (Figure 4I–L). Moreover, we showed spatial differences in the locations of Pax9 and Osr2 gene expression in the E13.5 and E14.5 palatal shelves.

### 3.4. Temporal Differences in the Gene Expression Locations of Molecules

To clarify the temporal differences, we next compared the gene expression levels in E13.5 and E14.5 in the lingual/nasal region of the palatal shelf (Figure 5A–F). Pax9 gene expression was not significantly different between E13.5 and E14.5 (Figure 5A). Osr2 gene expression was significantly higher at E13.5 than at E14.5 in the anterior, middle, and posterior parts (Figure 5B). There were no significant differences in the gene expression of Tgfβ3, F-actin, E-cadherin, or Ki67 between E13.5 and E14.5 (Figure 5C–F).

In the buccal/oral region, a similar expression trend was observed for the lingual/nasal region. Osr2 gene expression was significantly higher at E13.5 than at E14.5 in the anterior, middle, and posterior parts (Figure 5H). There were no significant differences in the gene expression of Pax9, Tgfβ3, F-actin, E-cadherin, or Ki67 between E13.5 and E14.5 (Figure 5G,I–L). We showed temporal differences in the locations of Osr2 gene expression in the E13.5 and E14.5 palatal shelf.

## 4. Discussion

In this study, we investigated the changes in the spatiotemporal expression locations of molecules related to palatal shelf elevation and tissue deformation before and after elevation. Immunohistochemistry and RT–PCR focusing on three regions along the anterior–posterior axis of the palatal shelf showed significant changes in Pax9 and Osr2. Pax9 and Osr2 are important regulators of palatal mesenchyme proliferation and function, working in concert [12]. Pax9-mutant palatal shelves exhibited delayed elevation in the anterior and middle parts at E14.5. At E15.5, the palatal shelf was elevated but not elongated toward the midline, resulting in a cleft palate. In the posterior part of Pax9-mutant palatal shelves at E14.5, the vertically elongated shelf was shorter. At E15.5, palatal shelves were not observed [6,18]. Osr2-mutant mice showed cleft palate due to impaired mesenchymal cell proliferation and delayed palatal elevation, and the same was true for Pax9-mutant mice [15]. This dysplasia of the palatal shelves in these mutant mice is suggested to be due to decreased mesenchymal cell proliferation. Pax9 was more strongly expressed in the lingual/nasal region than in the buccal/oral region in the anterior and middle parts of the palatal mesenchyme at E13.5. In contrast, in the posterior part of the palatal mesenchyme between E13.5 and E14.5, Pax9 was more strongly expressed in the buccal/oral region than in the lingual/nasal region (Figure 4A,G). Moreover, we showed that Osr2 gene expression was significantly higher in the lingual/nasal region than in the buccal/nasal region in E13.5 (Figure 4B), whereas in E14.5, Osr2 expression was reduced throughout the palatal shelf from the anterior to the posterior region compared to E13.5. This spatiotemporal difference in the expression region may be one of the factors leading to a bias in cell behavior, including cell number and cell density, leading to differences in the elevation pattern (“flip-up” model and “flow” model) in normal palatal development. Meanwhile, we showed that Ki67 was widely expressed in the epithelium and mesenchyme from the anterior to posterior palatal shelf (Figure 3M–R). It has been reported that cell proliferation is not biased in the palatal mesenchymal cells, which is consistent with our results [26]. However, since our results were obtained before and after elevation time points, cell proliferation and density in mesenchymal cells may be biased immediately before and during elevation.

A recent study reported that Osr2 negatively regulates the Sema3a and Sema3d genes, which encode members of class 3 semaphorins [27], which are expressed in the developing palatal mesenchyme [28]. The Sema3a and Sema3d mRNAs were expressed in the anterior to posterior regions in the lingual/nasal regions at E13.5 [28] and not in the Osr2 expression regions (the buccal/oral region from anterior to posterior parts) obtained in this study (Figure 4B). In addition, the above report suggested that Osr2 contributes to normal palate development via Sema3 signaling, since the Sema3 signaling family plays a role in regulating cell proliferation, migration, and differentiation [28]. The influence of cell migration on palatal elevation is still unknown, but cell migration is an important factor in tissue formation. Cell migration in the lingual/nasal regions where Sema3a and Sema3d are expressed may contribute to normal palatal development, including elevation.

Tgfβ3 was expressed at both E13.5 and E14.5 in the anterior-to-posterior epithelium, and it was equally expressed in the lingual/nasal and buccal/oral regions (Figure 2M–R). Tgfβ3 has been reported to be a key factor in removing epithelial cells called medial edge epithelial cells and facilitating the disappearance of the midline epithelial seam during palatal fusion [10,29]. Mutant mice lacking Tgfβ3 exhibited a cleft palate [14], and Pax9 expression was reduced in the anterior-to-posterior regions at the palatal medial edge [19]. In addition, it was reported that Tgfβ increases interferon regulatory factor 6 (IRF6) expression, suggesting that IRF6 is important for palatal development [30]. Another report suggested that IRF6 may regulate cell migration [31]. Whether and how Tgfβ3 is involved in normal palate elevation is unknown. Nevertheless, the observation of Tgfβ3 expression at E13.5 before elevation (Figure 2M–O) suggests that Tgfβ3, along with Pax9 and IRF6, is involved in the behavior that contributes to palatal elevation in mesenchymal and epithelial cells.

Palatal elevation is a large-scale tissue deformation that is thought to occur and be accomplished by the coordination of mechanical factors at the cellular and tissue levels. Although the underlying mechanism of this deformation is not understood, remodeling of the extracellular matrix and cytoskeletons, local changes in cell density, and cell proliferation have been hypothesized to be driving forces behind the elevation [15,26,32,33,34]. F-actin was expressed throughout the epithelium and mesenchyme, while E-cadherin was only expressed in the epithelium. These were similarly expressed from the anterior to posterior parts in the lingual/nasal and buccal/oral regions at E13.5 and E14.5 (Figure 3A–L). It has been suggested that actin-dependent contractility in the palatal mesenchyme may contribute to elevation by changing the arrangement of mesenchymal cells [32], and actomyosin-driven cellular extrusion is integral to palatal fusion [34]. In addition, forces generated in the epithelium by actomyosin are known to be important in tissue deformation [35,36]. On the other hand, E-cadherin is thought to sustain tissue structure through cell adhesion, and it was reported that E-cadherin is a mechanosensor mediated by actomyosin that probes the mechanical environment [37]. Actin and E-cadherin probably work in concert to regulate the magnitude and direction of the generated forces, thereby contributing to the formation and sustainment of a normal palate.

We examined the changes in the spatiotemporal localization of molecules expressed in the entire palatal tissue, including the epithelium and mesenchyme, after and before elevation. No change in expression location was observed for some molecules, but since elevation is thought to be elevated for a short time period, observations on shorter time scales may bias the locational distribution of the expressed molecules. In addition, more detailed observation of cell behavior is expected to lead to an understanding of palatal elevation, including differences in elevation patterns in the anterior–posterior axis.

## Figures and Tables

**Figure 1 ijms-23-05160-f001:**
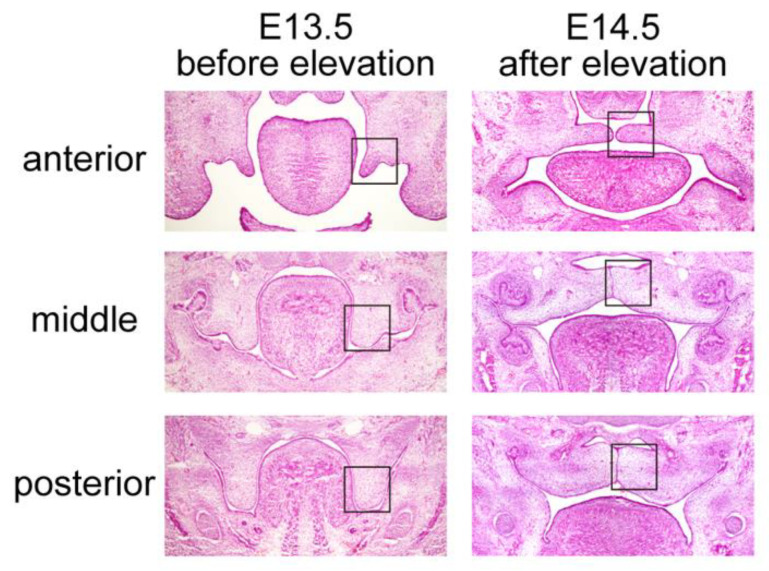
Hematoxylin- and eosin-stained coronal sections of the developing palatal shelves. Histological changes before (E13.5) and after (E14.5) palatal elevation. Coronal planes along the anterior–posterior axis in embryos: anterior (anterior to molar tooth bud), middle (same plane as molar tooth bud), and posterior (posterior to molar tooth bud). The boxed areas show the corresponding areas at higher magnification in Figure 2 and Figure 3. Scale bar: 500 μm.

**Figure 2 ijms-23-05160-f002:**
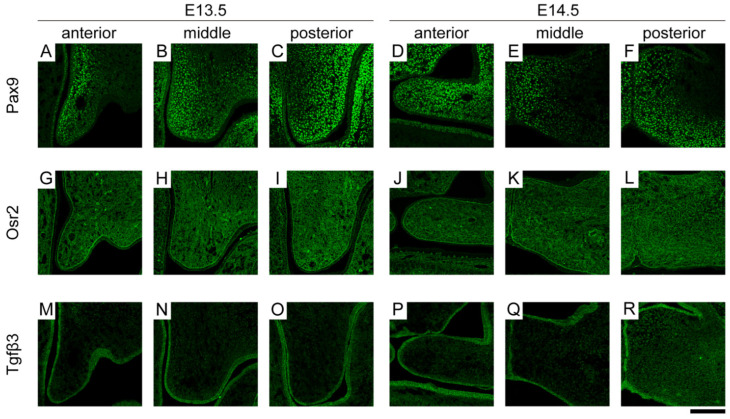
The immunohistochemical patterns of Pax9, Osr2, and Tgfβ3 expression during palate development. Confocal micrographs of immunostaining for (**A**–**F**) Pax9, (**G**–**L**) Osr2, and (**M**–**R**) Tgfβ3 obtained at both E13.5 and E14.5 palatal shelves in anterior-to-posterior parts. Scale bar: 50 μm.

**Figure 3 ijms-23-05160-f003:**
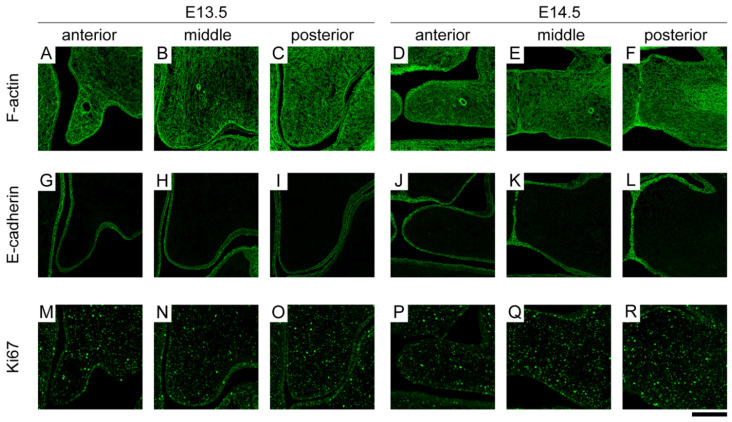
The immunohistochemical patterns of F-actin, E-cadherin, and Ki67 expression during palate development. Confocal micrographs of immunostaining for (**A**–**F**) F-actin, (**G**–**L**) E-cadherin, and (**M**–**R**) Ki67 obtained at both E13.5 and E14.5 palatal shelves in anterior-to-posterior parts. Scale bar: 50 μm.

**Figure 4 ijms-23-05160-f004:**
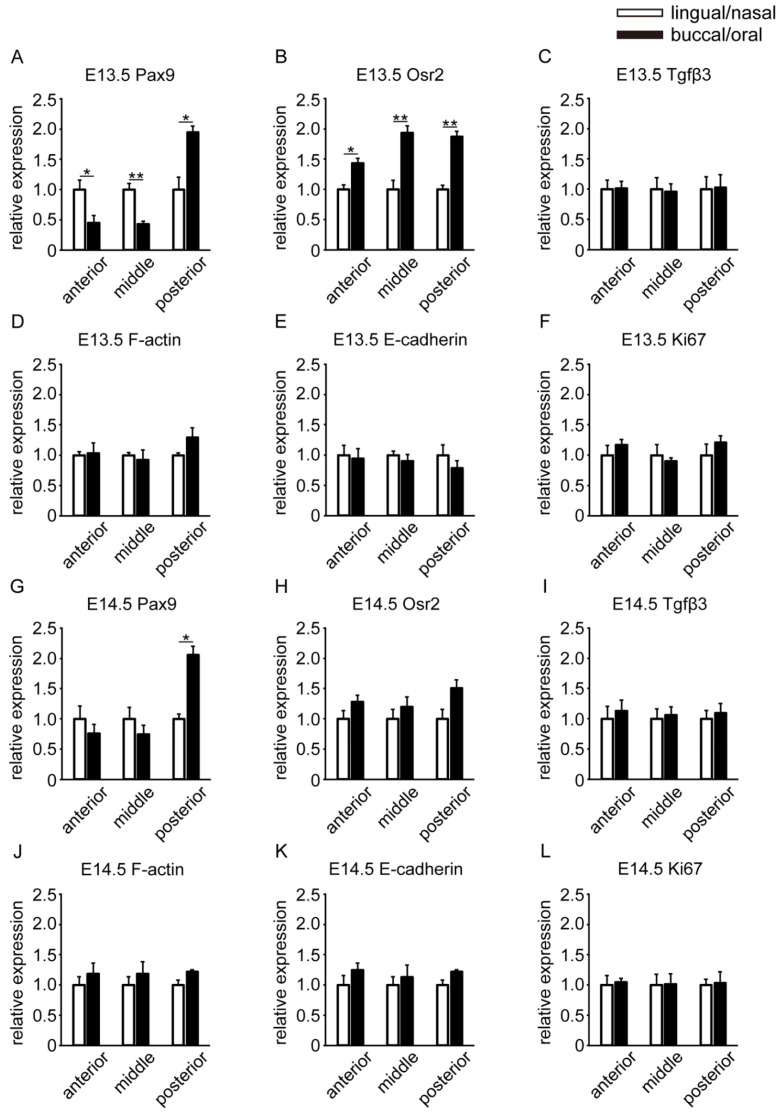
Spatial differences in anterior-to-posterior parts of the palatal shelf in both (**A**–**F**) E13.5 and (**G**–**L**) E14.5 through an RT–PCR assay. The results are expressed as the fold change in gene expression compared to lingual/nasal regions as a control and are presented as the mean ± SEM (*n* = 5). Student’s *t*-test was used to analyze the difference. * *p* < 0.05 and ** *p* < 0.01 were considered statistically significant.

**Figure 5 ijms-23-05160-f005:**
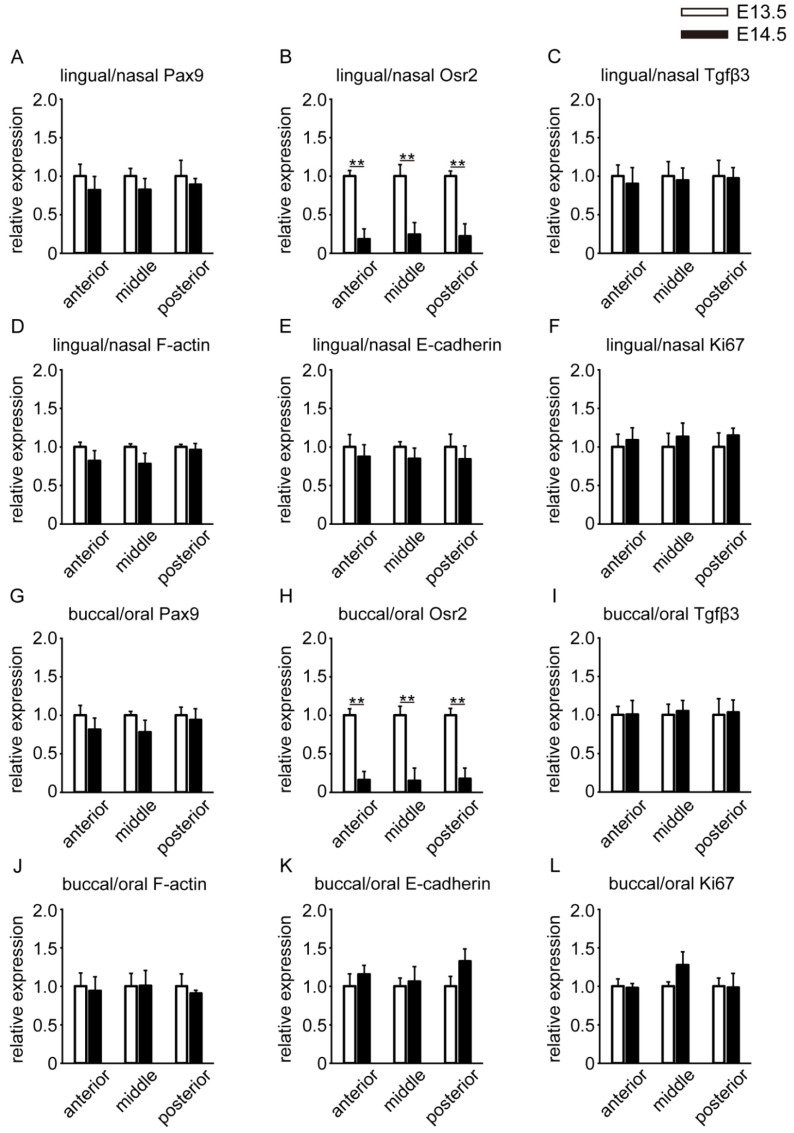
Temporal differences in anterior-to-posterior parts of the palatal shelf in both (**A**–**F**) lingual/nasal and (**G**–**L**) buccal/oral regions through an RT–PCR assay. The results are expressed as the fold change in gene expression compared to E13.5 as a control and are presented as the mean ± SEM (*n* = 5). Student’s *t*-test was used to analyze the difference. ** *p* < 0.01 was considered statistically significant.

## Data Availability

Not applicable.

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
