# Peer review of "Spatiotemporal Gene Expression Regions along the Anterior–Posterior Axis in Mouse Embryos before and after Palatal Elevation"

_ijms, 2022, doi:10.3390/ijms23095160_

Round 1

Reviewer 1 Report

In the present study, it is not clear what the hypothesis or research question driving the research were, as these are not clearly stated in the introductory or concluding statements of the article.

Furthermore, there are some concerns with the lack of novelty of this study... For example, the temporospatial expression of Pax9, Osr2 and Tgf-B using IHC and RT-qPCR methods has already been well defined in the present literature: 

Jia S, Zhou J, D'Souza RN. Pax9's dual roles in modulating Wnt signaling during murine palatogenesis. Dev Dyn. 2020 Oct;249(10):1274-1284. doi: 10.1002/dvdy.189. Epub 2020 Aug 4. PMID: 32390226.

Li R, Chen Z, Yu Q, Weng M, Chen Z. The Function and Regulatory Network of Pax9 Gene in Palate Development. J Dent Res. 2019 Mar;98(3):277-287. doi: 10.1177/0022034518811861. Epub 2018 Dec 24. PMID: 30583699.

Also, these above cited studies included additional research design which is lacking in the present study here;  The authors should consider expanding the robustness of their dataset using additional approaches to quantify the spatiotemporal expression profiles of these critical molecules driving palate formation. 

In its present form, this article lack novelty and robustness to be included a substantial advancement in the field. 

Author Response

We appreciate the reviewer’s critical comment. The purpose of this study was to elucidate the spatial and temporal expression patterns of molecule related to palatal shelf elevation and tissue deformation before and after elevation during mouse palatogenesis. To perform spatial analysis, we examined the lingual/nasal and buccal/oral regions of the palatal shelf separately for the three parts along the anterior-posterior axis. Moreover, to perform temporal analysis, we examined E13.5 and E14.5 mouse embryos. As the reviewer suggested, previous study showed the spatiotemporal expression of Pax9, Osr2, and Tgfβ3 in the paper you introduced us to. However, those data cannot be simply compared due to differences in the genetic backgrounds and the lack of analysis regions. To reveal differences in spatiotemporal expression pattern, it is essential to compare them under the same condition. Therefore, we examined the expression pattern over a wide area of the palatal shelf before and after elevation using same genetic background mouse. As a result, significant expression changes were observed, especially in Pax9 and Osr2. We added description it in the introduction section and discussion section.

Page 2, lines 59-70:

These different elevation patterns suggest the involvement of region-specific molecules. For example, the expression pattern of Pax9, Osr2 and Tgfβ have been reported by in situ hybridization analysis [15,18-20]. However, those data cannot be simply compared due to differences in the genetic backgrounds and the lack of analysis regions. Therefore, we examined the expression region of molecules associated with palatal shelf elevation and tissue deformation in mouse embryos with same genetic background. To perform more detailed spatiotemporal analysis, we examined the lingual/nasal and buccal/oral regions of the palatal shelf in E13.5 and E14.5 mouse embryos before and after elevation separately for the three parts along the anterior-posterior axis of the palate: anterior, middle, and posterior. In each of these regions, we examined the expression regions of Pax9, Osr2, and Tgfβ3 related to palatal shelf elevation and F-actin, E-cadherin, and Ki67 related to tissue deformation.

Page 9, lines 215-218:

In this study, we investigated the changes in the spatiotemporal expression locations of molecules related to palatal shelf elevation and tissue deformation before and after elevation. Immunohistochemistry and RT–PCR focusing on three regions along the anterior-posterior axis of the palatal shelf showed significant changes in Pax9 and Osr2.

Reviewer 2 Report

Congratulations to Authors for this very interesting manuscript. It shows a good appropriateness of topic for IJMS journal, good research design and methodological soundness. The statistics are appropriate for the study. The findings, discussion and conclusions are good. I suggest to publish  the manuscript.  

Author Response

We thank you for your peer review.

Reviewer 3 Report

  1. The paper in the introduction section should have the information, that the PAX 9 could be responsible for hypodontia (and with gene MSX1) is both responsible for clefts and lacking tooth buds as well - please, find the proper references.
  2. Please add more information pn Pax9, Osr2 and Tgfβ3 expression during palate development at the introduction section (it would be valid to add a table compating those genes).
  3. Please, state if you have the permission for the bioethics committee (or it is not required in Japan) in the materials and methods secion
  4. The English is understandable to me, but I am not a Native Speaker, therefore I will not be able to judge this
  5. Would you please be able to make a graph of palatal elevation, so that it would be clear to visualize this.

I would like to thank the Authors for the opportunity to revise this paper. It is definately valid publishing within the highly rated journal. I wish you the very best, please just note those of my suggestions. The paper is full of figures, which makes this paper even more worth and interesting.

Round 2

Reviewer 1 Report

I thank the authors for their efforts in responding to the feedback presented to them from all reviewers.  While there remain some critical gaps in the present study, if the authors are willing to directly address these gaps as previously described in the first round of reviewer feedback, then the scientific soundness would be sufficient to publish in IJMS.  Furthermore, if the authors could please pass their entire manuscript through strict English editorial services (beyond what was previously done) then this would help in the presentation of this data to the scientific community. 

Author Response

We appreciate the reviewer’s comment. This paper edited by MDPI Language Editing Service, and provided the Editing Certificate in cover letter.